# Outcomes of asymptomatic recombinant tissue plasminogen activator associated intracranial hemorrhage

**Chutithep Teekaput**[1], **Kitti Thiankhaw**[1], **Surat Tanprawate**[1], **Kanokkarn Teekaput**[1], **Chatree Chai-Adisaksopha**[2]*

1 Division of Neurology, Department of Internal Medicine, Faculty of Medicine, Chiang Mai University, Chiang Mai, Thailand, 2 Division of Hematology, Department of Internal Medicine, Faculty of Medicine, Chiang Mai University, Chiang Mai, Thailand

* chatree.chai@cmu.ac.th

## Abstract

### Background

Intracranial hemorrhage (ICH) is the most devastating complication of recombinant tissue plasminogen activator (rtPA) treatment in acute ischemic stroke patients. Data on rtPA-associated asymptomatic ICH (aICH) are limited.

### Objectives

To determine the incidence, risk factors, characteristics, management, and clinical outcome of rtPA-associated aICH.

### Methods

The data were retrieved from the Chiang Mai University Hospital Stroke Registry between 1995 and 2019. Consecutive ischemic stroke patients were included if they were 18 or older and received rtPA. Study outcomes were the incidence and characteristics of aICH, management, 90-day modified Rankin scale (mRS), National Institute of Health Stroke Scale (NIHSS), Barthel index, and all-cause mortality.

### Results

Of 725 rtPA treated patients, 166 (16.0%, 95% confidence interval [CI] 13.4–18.9) had aICH, 50 (6.9%, 95% CI 5.2–9.0) had symptomatic ICH (sICH). Patients with aICH had more hemorrhagic infarctions (HI) compared to sICH (81.9% vs 2.0%, P-value < 0.001). Fresh Frozen Plasma and cryoprecipitate were the most common blood products used to reverse the anticoagulant effect in sICH. Craniotomy was performed in 1% and 60% of patients who had aICH and sICH. At 90 days, patients who had aICH had poorer clinical outcomes (mRS, NIHSS and Barthel index) as compared to those without ICH. Compared to non-ICH patients, aICH patients were associated with increased risk of 90-day mortality, the hazard ratio (HR), 3.7, 95% CI 1.6–8.9.

**Data Availability Statement:** Data cannot be shared publicly because of ethical issues. Data are available from the Ethics Committee of Faculty of Medicine, Chiang Mai University (contact via

Researchmed@cmu.ac.th) for researchers who meet the criteria for access to confidential data.

**Funding:** The authors received no specific funding for this work.

**Competing interests:** The authors have declared that no competing interests exist.

## Conclusions

The rtPA-associated aICH increased the risk of morbidity and mortality outcomes. Further treatment consensus, guideline generation, or clinical trials focusing on the treatment of rtPA-associated aICH may be required.

## Introduction

Symptomatic intracranial hemorrhage (sICH) after thrombolytic treatment in acute stroke is one of the most devastating complications in the recombinant tissue plasminogen activator (rtPA) era. The rate of sICH was reported to range from two to seven percent. Most of the events develop within 36 hours after infusion [1, 2]. According to the European Cooperative Acute Stroke Study (ECASS), sICH cases are classified into two major groups: hemorrhagic infarction without mass effect (HI) and parenchymal hematoma with mass effect (PH) [3, 4]. The PH group has a significant space-occupying impact, which carried 50% of mortality and morbidity [5]. The aims of sICH treatment consist of three major modalities 1) reversal of coagulopathy with various agents, 2) hematoma expansion prevention, and 3) neurosurgical intervention.

Although numerous research has focused on the long-term outcomes of sICH, there is insufficient evidence on the long-term outcomes of asymptomatic ICH (aICH). Moreover, there is no consensus or standard recommendation for medical and surgical management of rtPA-associated intracranial bleeding, especially in aICH patients.

The objective of this study is to determine the incidence, risk factors, and characteristics of aICH in stroke patients after receiving thrombolytic therapy. We also determined the reversal strategies, surgical intervention, and the outcomes.

## Methods

Data were retrieved from the Chiang Mai University Hospital Stroke Registry between 1995 and 2019. This registry prospectively collected consecutive patients who were diagnosed with all types of acute stroke. All stroke patients who were 18 years or older, diagnosed with acute ischemic stroke with onset less than 3–4.5 hours, without contraindications to rtPA administration (e.g. history of previous intracranial hemorrhage or abnormal laboratory, etc.) would be given rtPA 0.9 mg/kg, maximum dose 90 mg over 60 minutes intravenously within 60 minutes after arrival at an emergency department. The clinical data, laboratory, imaging, and outcomes were systematically collected. All patients were subsequently admitted to the Acute Stroke Unit (ASU). Vital signs and neurological outcomes including the National Institute of Health Stroke Scale (NIHSS), modified Rankin scale (mRS), and Barthel index were monitored. All patients underwent a follow-up computed tomography (CT) scan of the brain 24 hours after receiving rtPA to detect the radiologic signs of intracranial bleeding. In case of worsening neurological symptoms, defined as an increase of NIHSS $\geq$ 4, the patients would undergo emergency brain imaging. If intracranial bleeding was detected, the comprehensive ASU care team comprised of neurologists, neurosurgeons, rehabilitation physicians, nurses, and pharmacists would evaluate and classify patients into symptomatic or asymptomatic intracranial hemorrhage. Neurosurgical intervention or reversal agents would be given according to the comprehensive ASU care team consensus. When patients were discharged from the ASU, NIHSS, mRS, Barthel index and all-cause mortality were recorded. The NIHSS, mRS, Barthel index, and all-cause mortality would be re-collected at the 14-day and 90-day follow-

up visits. Patients who did not have a follow-up brain imaging or who had missing data on functional outcomes at any time point were excluded from the study.

We analyze data from the registry from three time points: at diagnosis, 14 days and 90 days after onset. At the time of stroke diagnosis, we collected demographic data, medical history, laboratory data, and imaging data. Bleeding events were collected and classified into intracranial (sICH or aICH) or extracranial sites. The all-cause mortality rate was evaluated at 14-day and 90-day post-rtPA administration for all patients. Among patients who experienced rtPA bleeding, data were collected regarding reversal agents, blood product transfusion, and surgical intervention. The follow-up imaging results at the time of discharge, 14 days, and 90 days after rtPA treatment were also recorded. Functional outcomes, including mRS, NIHSS, and Barthel Index were collected at the time of stroke diagnosis, 14 days, and 90 days post-rtPA.

A primary outcome of this study was 90-day mRS. The mRS was stratified in seven levels from zero to six. The higher mRS reflected the poorer neurological outcomes [6]. The secondary outcomes were the incidence of sICH, aICH, extracranial bleeding, clinical outcomes following intracranial hemorrhage (ICH), NIHSS, Barthel index and all-cause mortality, types, and frequency of reversal agents and neurosurgical intervention.

Clinical data were presented in numbers and percent (%) and 95% confidence interval (CI) for categorical data. Mean and corresponding standard deviation (SD), or median and corresponding interquartile range (IQR) were reported for continuous data as appropriate. Comparisons of demographic data and clinical characteristics between groups were performed using Student's t-test, Mann-Whitney U test, Chi-square test, or Fisher's exact test as appropriate.

The univariable analysis was performed to determine the risk factors of ICH using logistic regression. Variables with P-value < 0.10 from the univariable analysis were tested in multi-variable models. A two-sided test at a P-value of < 0.05 was used to indicate statistical significance. Kaplan-Meier curves were generated to assess the survival rate among patients. Cox-Proportional Hazard (PH) model was used to determine the hazard ratio among groups. All statistical analyses were performed using Stata statistical software version 16.1 (Stata Statistical Software: Release 16.1, Stata Corporation, College Station, TX, 2019).

The study received approval from the Research Ethics Committee, Faculty of Medicine, Chiang Mai University, Chiang Mai, 50200, Thailand. All data from this study were fully anonymized before access and the Research Ethics Committee waived the requirement for informed consent.

## Results

### Incidence and risk factors of ICH

Of 745 patients with acute ischemic stroke who received rtPA, 20 patients were excluded from this study due to stroke mimicker and incomplete data. Of the remaining 725 patients, 197 patients (27.2%, 95% confidence interval [CI] 23.9–30.6) were diagnosed with rtPA-associated hemorrhage at any sites. Total 166 patients (22.9% 95% CI 19.9–26.1) experienced ICH. Of these, 116 had aICH (16.0%, 95% CI 13.4–18.9) and 50 had sICH (6.9%, 95% CI 5.2–9.0). Thirty-one patients (4.3%, 95% CI 2.9–6.9) suffered an extracranial hemorrhage, see Fig 1.

Table 1 demonstrates the baseline clinical characteristics of patients, classified into three groups (no ICH, aICH, and sICH), see Table 1. Atrial fibrillation and prior antiplatelet use were found less frequently in the no ICH group compared to aICH and sICH groups. As compared to the no ICH group, the aICH and sICH groups had more unfavorable admission characteristics (higher mRS, higher NIHSS and lower Barthel index).

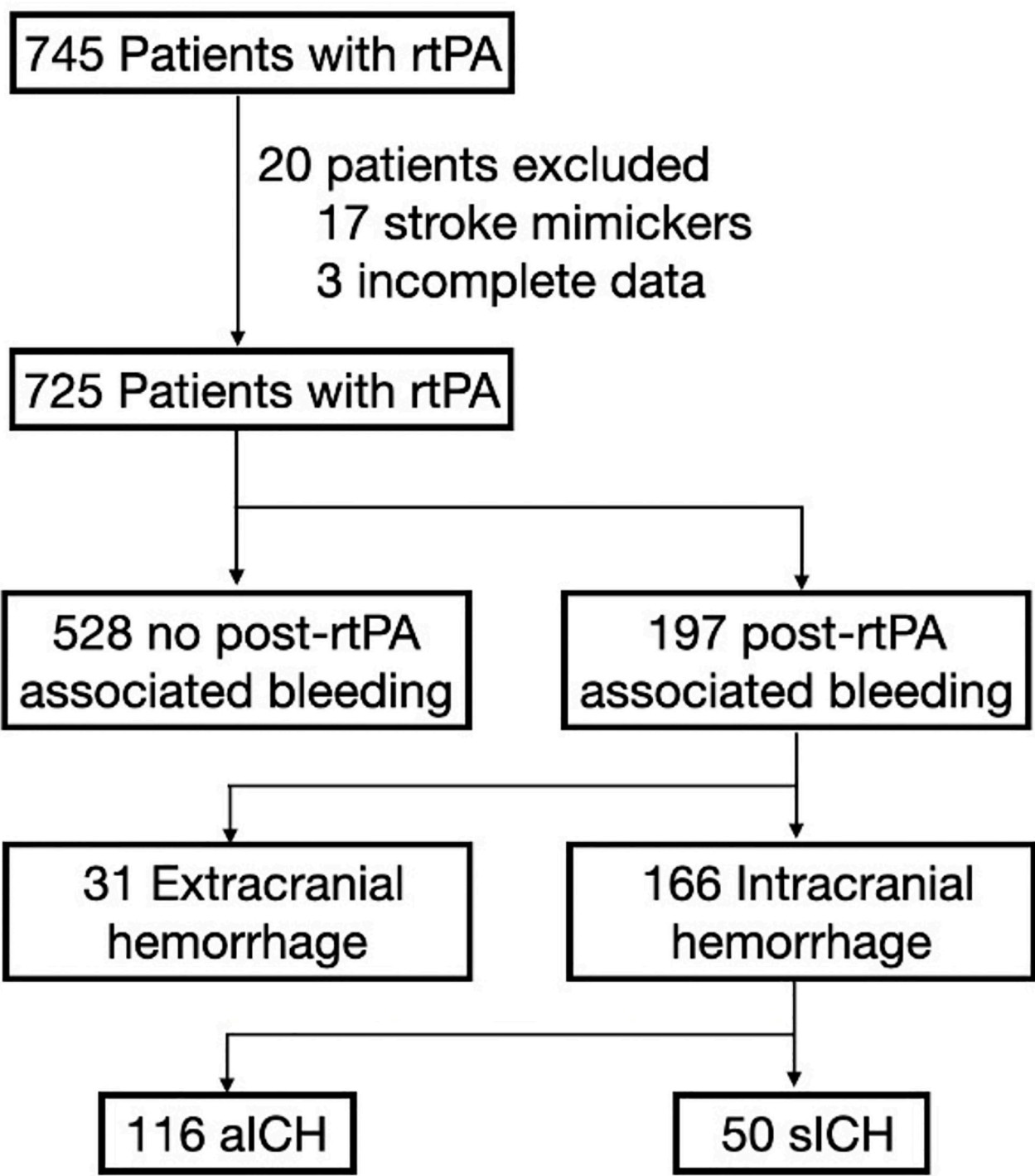

**Fig 1. Study flow diagram.** Abbreviation: rtPA, recombinant tissue plasminogen activator; aICH, asymptomatic intracranial hemorrhage; sICH, symptomatic intracranial hemorrhage.

**Table 1. Baseline characteristics of patients who received recombinant tissue plasminogen activator for acute ischemic stroke.**

| Characteristics/factors | No ICH (n = 559) | aICH (n = 116) | sICH (n = 50) | P-value | | |
|---|---|---|---|---|---|---|
| | | | | No ICH vs aICH | No ICH va sICH | aICH vs sICH |
| **Male sex—no. (%)** | 288 (51.5) | 51 (44.0) | 25 (50.0) | 0.14 | 0.84 | 0.47 |
| **Age—median (IQR)** | 65.0 (55.0, 75.0) | 67.0 (58.0, 76.0) | 70.5 (63.0, 74.0) | 0.37 | 0.87 | 0.63 |
| **Weight (kg)—median (IQR)** | 58.0 (50.7, 67.0) | 60.5 (50.0, 67.5) | 54.5 (48.0, 64.1) | 0.55 | 0.09 | 0.06 |
| **Co-morbid diseases—no. (%)** | | | | | | |
| **Hypertension** | 356 (63.7) | 82 (70.7) | 33 (66.0) | 0.15 | 0.74 | 0.55 |
| **Diabete mellitus** | 109 (19.5) | 27 (23.3) | 17 (34.0) | 0.36 | 0.02† | 0.15 |
| **Atrial fibrillation** | 67 (12.0) | 32 (27.6) | 19 (38.0) | <0.001† | <0.001† | 0.18 |
| **Dyslipidemia** | 170 (30.4) | 37 (31.9) | 14 (28.0) | 0.75 | 0.72 | 0.62 |
| **Congestive heart failure** | 39 (7.0) | 7 (6.0) | 7 (14.0) | 0.71 | 0.07 | 0.09 |
| **Coronary artery disease** | 52 (9.3) | 17 (14.7) | 6 (12.0) | 0.08 | 0.53 | 0.65 |
| **Peripheral arterial disease** | 2 (0.4) | 1 (0.9) | 0 (0.0) | 0.46 | 0.67 | 0.51 |
| **Recurrent stroke** | 44 (7.9) | 7 (6.0) | 2 (4.0) | 0.50 | 0.32 | 0.60 |
| **Smoking** | 79 (14.1) | 12 (10.3) | 8 (16.0) | 0.28 | 0.72 | 0.30 |
| **Alcohol consumption** | 57 (10.2) | 14 (12.1) | 9 (18.0) | 0.55 | 0.09 | 0.31 |
| **Antiplatelet/anticoagulant—no. (%)** | | | | | | |
| **Anticoagulant** | 33 (5.9) | 10 (8.6) | 3 (6.0) | 0.28 | 0.98 | 0.56 |
| **Antiplatelet** | 91 (16.3) | 31 (26.7) | 18 (36.0) | 0.01† | <0.001† | 0.23 |
| **TOAST classification—no. (%)** | | | | | | |
| **Atherosclerosis** | 331 (59.2) | 36 (31.0) | 17 (34.0) | <0.001† | 0.001† | 0.71 |
| **Cardioembolism** | 150 (26.8) | 70 (60.3) | 31 (62.0) | <0.001† | <0.001† | 0.84 |
| **Small vessel diseases** | 48 (8.6) | 5 (4.3) | 1 (2.0) | 0.12 | 0.10 | 0.46 |
| **Other etiology** | 17 (3.0) | 5 (4.3) | 1 (2.0) | 0.48 | 0.68 | 0.46 |
| **Undetermined** | 13 (2.3) | 0 (0.0) | 0 (0.0) | 0.10 | 0.28 | - |
| **Admission profiles—median (IQR)** | | | | | | |
| **Onset to needle (minutes)** | 159.0 (122.0, 200.0) | 154.5 (120.0, 186.0) | 150,0 (120.0, 210.0) | 0.15 | 0.52 | 0.78 |
| **Door to needle (minutes)** | 55.0 (45.0, 71.0) | 55.0 (46.0, 66.0) | 50.0 (44.0, 66.0) | 0.15 | 0.82 | 0.48 |
| **Admission NIHSS—median (IQR)** | 9 (5, 15) | 13 (8, 18) | 17 (12, 21) | 0.05† | <0.001† | 0.21 |
| **Admission Barthel index—median (IQR)** | 25 (10, 55) | 20 (5, 40) | 10 (0, 15) | 0.03† | 0.01† | 0.22 |
| **Admission mRS >2—no. (%)** | 514 (92.0) | 112 (96.6) | 49 (98.0) | 0.08 | 0.12 | 0.62 |

Abbreviations: ICH, intracranial hemorrhage; aICH, asymptomatic intracranial hemorrhage; sICH, symptomatic intracranial hemorrhage; rtPA, recombinant tissue plasminogen activator; kg, kilogram; TOAST, the trial of ORG 10172 in acute stroke treatment; IQR, interquartile range; SD, standard deviation; NIHSS, National Institute of Health Stroke Scale; mRS, modified Rankin Scale

†statistically significant

### Risk factors of asymptomatic intracranial hemorrhage

Table 2 demonstrates the results of a multivariable analysis of risk factors for aICH, see Table 2. The cardioembolic stroke subtype was associated with an increased risk of aICH with an OR of 3.7 (95% CI 2.4–5.7). Moreover, admission NIHSS of more than 15 was associated with the increment of aICH, OR 1.9 (95% CI 1.2–3.0).

### Characteristics of rtPA associated intracranial hemorrhage

Table 3 demonstrates the characteristics of rtPA-associated intracranial hemorrhage, see Table 3. Intracranial hemorrhage occurring less than 24 hours after rtPA administration was reported in 5.2% of patients with aICH and 34.0% of those with sICH (P-value <0.001).

**Table 2. Risk factors of asymptomatic intracranial hemorrhage.**

| Risk factor | Odds Ratios (95% Confident Interval) | P-value |
|---|---|---|
| Cardioembolic stroke subtype | 3.7 (2.4–5.7) | <0.001 |
| Admission NIHSS >15 | 1.9 (1.2–3.0) | 0.003 |

Abbreviations: NIHSS, National Institute of Health Stroke Scale

Intracranial bleeding, which primarily developed within 48 hours after rtPA administration, was reported in 78.5% in the aICH group and 92.0% in the sICH group, P-value = 0.04. According to the National Institute of Neurological Diseases and Stroke (NINDS) classification, patients with aICH had more hemorrhagic infarctions (HI) compared to sICH (81.9% vs 2.0%, P-value <0.001). Contrarily, patients with aICH were less likely to have parenchymal hemorrhage (PH) compared to sICH (18.1% vs 98.0%, P-value <0.001). Hypofibrinogenemia, defined as fibrinogen < 100 mg/dL, was less common in aICH group comparing to sICH group (1.7% vs 14.0%, P-value <0.001).

## Management of rtPA associated intracranial hemorrhage

Table 4 demonstrates the management of rtPA-associated bleeding, see Table 4. According to the lower incidence of bleeding progression in the aICH group, patients with aICH received fewer blood products than patients with sICH (7.8% vs 80.0 percent, P-value <0.001). Fresh

**Table 3. Characteristics of rtPA associated intracranial hemorrhage.**

| Characteristics | Asymptomatic intracranial hemorrhage (n = 116) | Symptomatic intracranial hemorrhage (n = 50) | P-value |
|---|---|---|---|
| **Onset to bleeding—no. (%)** | | | |
| less than 24 hours | 6 (5.2) | 17 (34.0) | <0.001† |
| less than 48 hours | 91 (78.5) | 46 (92.0) | 0.04† |
| after 3 days | 18 (15.5) | 2 (4.0) | 0.04† |
| Fibrinogen <100 (mg/dL)—no. (%) | 2 (1.7) | 7 (14.0) | <0.001† |
| **Bleeding characteristic—NINDS classification—no. (%)** | | | |
| Hemorrhagic infarction | 95 (81.9) | 1 (2.0) | <0.001† |
| Parenchymal hemorrhage | 21 (18.1) | 49 (98.0) | <0.001† |
| Bleeding progression—no. (%) | 12 (14.5) | 12 (37.5) | 0.01† |

Abbreviations: rtPA, recombinant tissue plasminogen activator; NINDS, National Institute of Neurological Diseases and Stroke; mg, milligram; dL, decilitre

†statistically significant

**Table 4. Frequency of reversal agents and neurosurgical intervention classified by asymptomatic intracranial bleeding and symptomatic intracranial hemorrhage.**

| Treatment—no. (%) | Asymptomatic intracranial hemorrhage (n = 116) | Symptomatic intracranial hemorrhage (n = 50) | P-value |
|---|---|---|---|
| Cryoprecipitate | 2 (1.7) | 17 (34.0) | <0.001† |
| FFP | 5 (4.3) | 31 (62.0) | <0.001† |
| Vitamin K | 0 (0.0) | 3 (6.0) | 0.03† |
| Platelet concentrate | 1 (0.9) | 10 (20.0) | <0.001† |
| Neurosurgery | 1 (0.9) | 30 (60.0) | <0.001† |
| PRC | 1 (0.9) | 16 (32.0) | <0.001† |

Abbreviations: FFP, fresh frozen plasma; PRC, packed red cells

†statistically significant

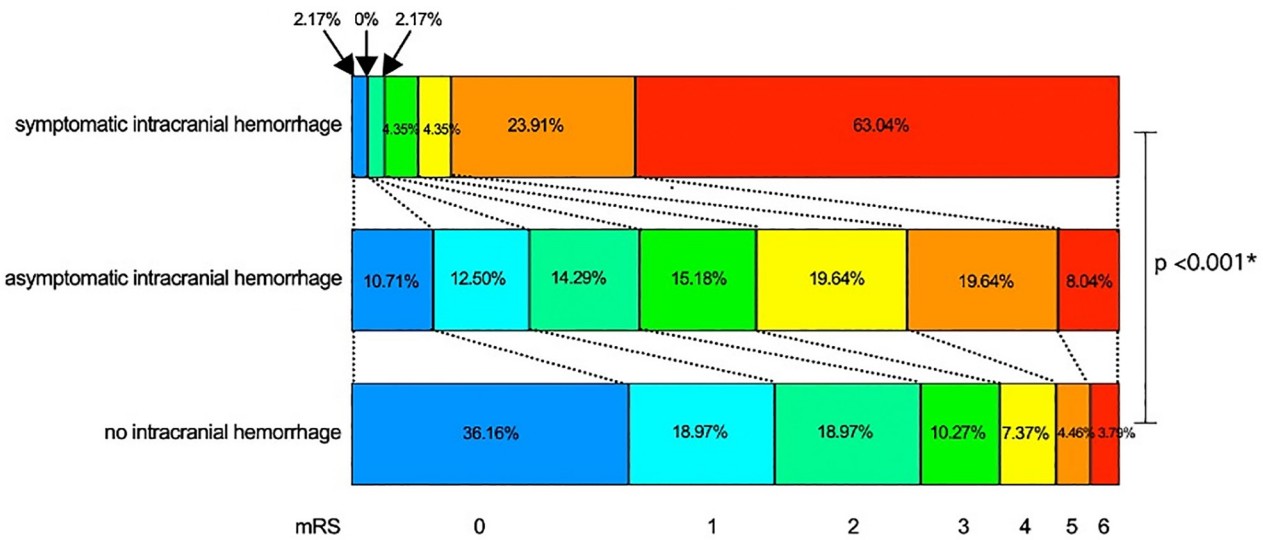

**Fig 2. 90-day modified Rankin scale in no intracranial bleeding, asymptomatic intracranial bleeding and symptomatic intracranial bleeding.** This figure demonstrated the mRS among sICH, aICH, and no intracranial hemorrhage. The unfavorable mRS (mRS >2) were found in sICH comparing with aICH and no bleeding group (96.0%, 63.8%, and 40.50% respectively, P-value <0.001) Abbreviations: mRS, modified Rankin Scale; sICH, symptomatic intracranial hemorrhage; aICH, asymptomatic intracranial hemorrhage.

frozen plasma followed by cryoprecipitate was the most common blood product used. Packed red blood cell transfusion was given in 0.9% of the aICH group and 32.0% in the sICH group (P-value <0.001). Craniotomy was performed in 0.9% of the aICH group and 60.0% in the sICH group (P-value <0.001).

Focusing on patients without ICH, transfusion of blood product or neurosurgical intervention was not required in all patients.

## The 90-day modified Rankin Scale (mRS)

Fig 2 demonstrates the 90-day mRS among no ICH, aICH, and sICH groups. A higher proportion of patients with unfavorable mRS (mRS >2) were observed in the aICH (63.8%) and sICH groups (96.0%) compared to the no ICH group (25.9%), P-value <0.001.

Table 5 demonstrates a multivariable analysis of unfavorable mRS at 90 days, see Table 5. After adjusting for confounders (age, atrial fibrillation, serum albumin level, admission NIHSS, and Barthel index), aICH and sICH increased the risk of unfavorable stroke outcome when compared to patients with no ICH with OR 2.3 (95% CI 1.5–3.6), P-value <0.001, and OR 21.8 (95% CI 5.1–92.4), P-value <0.001, respectively.

Furthermore, the no ICH group had a shorter length of stay (LOS) compared with aICH group and sICH group (median five days [IQR 3, 9],10 days [IQR 5, 20] and nine days [IQR 5, 15], P-value <0.001), respectively.

## Outcomes of rtPA associated intracranial hemorrhage

Among patients who underwent the follow-up CT scan of the brain, bleeding progression was lower in the aICH group compared to the sICH group (14.5% vs 37.5%, P-value <0.001). At discharge, 30.2% and 86.0% of patients in the aICH and sICH groups had NIHSS scale >15 (P-value <0.001), whereas 20.0% of patients with no ICH had NIHSS >15.

**Table 5. Multivariable analysis of unfavorable outcomes defined as modified Rankin scale > 2 in patients with acute ischemic stroke and received rtPA.**

| Characteristics | Odds Ratios (95% Confident Interval) | P-value |
|---|---|---|
| No ICH | Reference | Reference |
| aICH | 2.3 (1.5–3.6) | <0.001 |
| sICH | 21.8 (5.1–92.4) | <0.001 |
| Age | 1.0 (1.0–1.0) | 0.01 |
| Prior atrial fibrillation | 2.4 (1.6–3.7) | <0.001 |
| Recurrent stroke | 2.7 (1.5–4.9) | 0.001 |
| Higher admission NIHSS | 1.1 (1.1–1.1) | <0.001 |
| Higher admission mRS | 1.4 (1.2–1.6) | <0.001 |
| Lower admission Barthel index | 1.0 (1.0–1.0) | <0.001 |
| Lower albumin level | 0.6 (0.4–0.8) | <0.001 |

Abbreviations: rtPA, recombinant tissue plasminogen activator; ICH, intracranial hemorrhage; aICH, asymptomatic intracranial hemorrhage; sICH, symptomatic intracranial hemorrhage; NIHSS, National Institute of Health Stroke Scale; mRS, modified Rankin Scale.

NIHSS and Barthel index at 14 and 90 days after stroke diagnosis are shown in Figs 3 and 4. A higher proportion of patients with sICH had the worst stroke outcomes (defined as NIHSS >15 or Barthel index <25) compared to those with no ICH and aICH at day 14 (P-value = 0.003 and P-value <0.001, respectively), and at day 90 (P-value <0.001 and P-value <0.001, respectively).

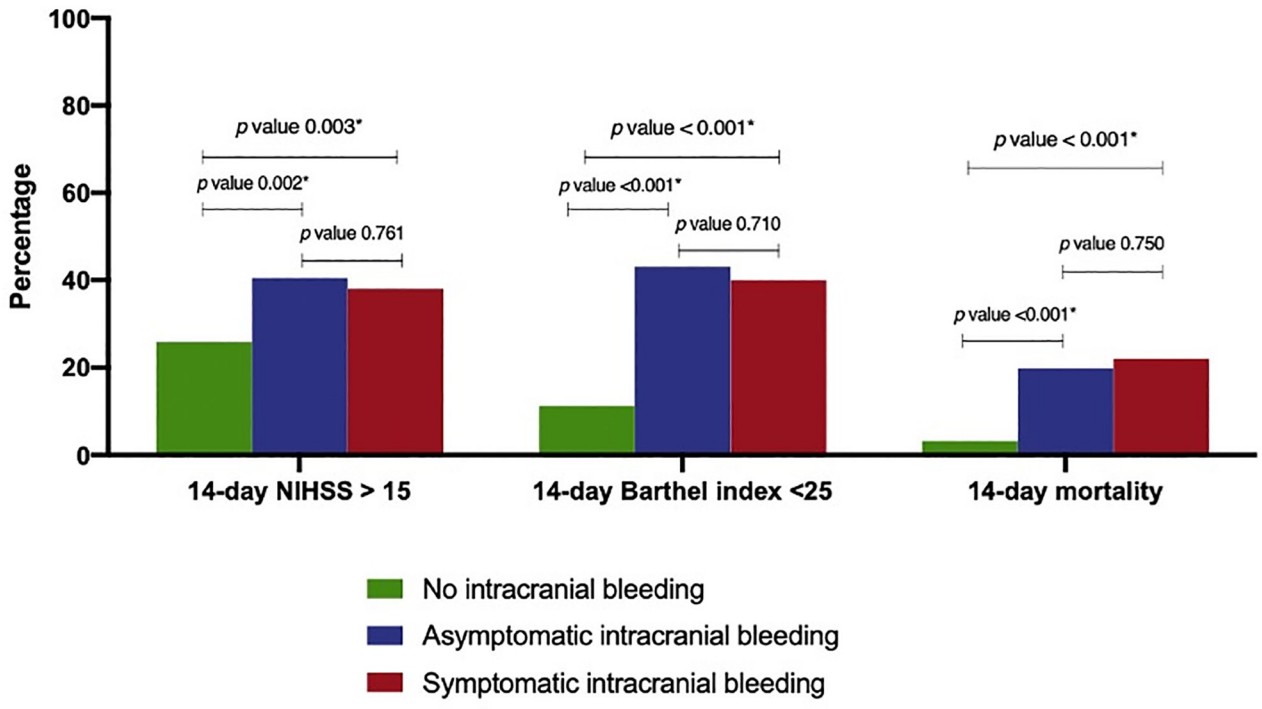

**Fig 3. 14-day outcomes (NIHSS >15, barthel index <25, mortality) in no intracranial hemorrhage, asymptomatic intracranial hemorrhage and symptomatic intracranial hemorrhage.** Abbreviations: NIHSS, National Institute of Health Stroke Scale.

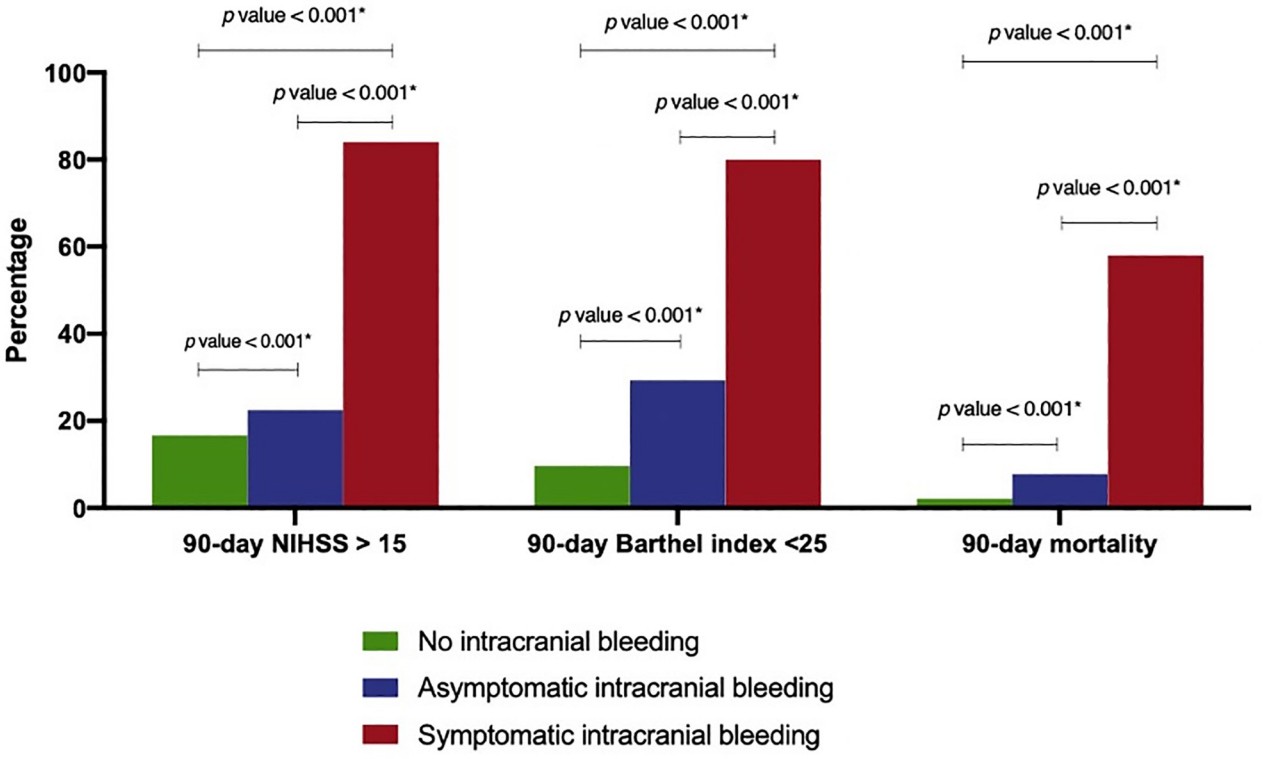

**Fig 4. 90-day outcomes (NIHSS >15, barthel index < 25, mortality) in no intracranial hemorrhage, asymptomatic intracranial hemorrhage, and symptomatic intracranial hemorrhage.** Abbreviations: NIHSS, National Institute of Health Stroke Scale.

Patients in the no ICH group had a significantly lower proportion of all-cause mortality at 14 days and 90 days as compared to patients with aICH and sICH. The Kaplan-Meier curve of overall survival is demonstrated in Fig 5. When compared to no ICH patients, aICH and sICH were associated with an increased risk of 90-day mortality adjusted by age, HR 3.7, 95% CI 1.6–8.9, and HR 40.8, 95% CI 20.7–80.6, respectively.

## Discussion

Even though the rtPA-associated intracranial hemorrhage is categorized into aICH and sICH, most of studies focus on the incidence, outcomes, and management only in patients suffering from sICH. According to the previous studies, the incidence of sICH after standard-dose rtPA treatment was ranging from two to seven percent [5]. The variations might be due to different definitions of sICH, populations, and research designs used in each study. In the ECASS (European Cooperative Acute Stroke Study) III, the incidence of sICH was reported higher in rtPA treatment compared with placebo (6.8% vs 1.3%, OR 5.55, 95% CI 4.01–7.70) [7]. According to our study, the incidence of sICH was similar to the results reported in the ECASS III study (6.9%, 95% CI 5.2–9.0).

Numerous rtPA-associated sICH factors have been reported. In one meta-analysis of 55 studies, older age, higher NIHSS, higher baseline glucose, hypertension, congestive heart failure, renal impairment, diabetes mellitus, ischemic heart disease, atrial fibrillation, and antiplatelet use were associated with an increased risk of sICH [8]. In our study, the embolic subtype according to the Trial of ORG 10172 in Acute Stroke Treatment (TOAST)

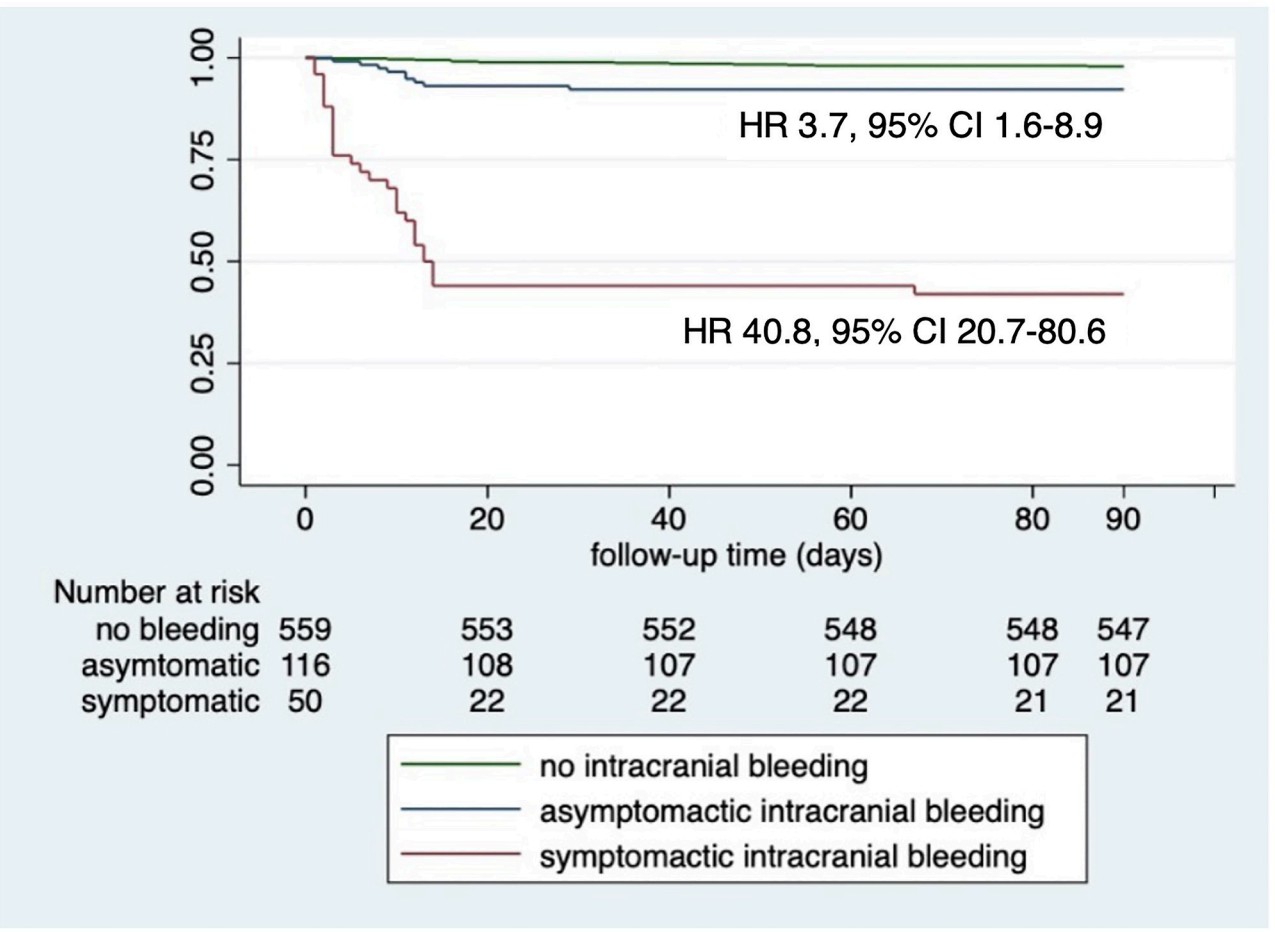

**Fig 5. The Kaplan-Meier survival curve among no intracranial hemorrhage, asymptomatic intracranial hemorrhage and symptomatic intracranial hemorrhage groups in 90 days follow-up after recombinant tissue Plasminogen Activator (rtPA).** When compared to non-ICH patients, asymptomatic ICH and symptomatic ICH was associated with an increased risk of 90-day mortality adjusted by age, HR 3.7, 95% CI 1.6–8.9, and HR 40.8, 95% CI 20.7–80.6, respectively." Abbreviations: ICH, intracranial hemorrhage; HR, Hazard Ratios; CI, confidence interval.

classification (Odds Ratios [OR] 3.4, 95% CI 2.4–5.0), concomitant antiplatelet usage (OR 1.8, 95% CI 1.2–2.8), higher NIHSS (OR 1.9, 95% CI 1.3–2.9), and lower Barthel index (OR 1.6, 95% CI 1.1–2.4) were found to be potential risks of rtPA associated bleeding. These minor variations might be from different study designs and sICH definitions used in individual studies.

Like an endogenous plasminogen activator, the rtPA achieves thrombolysis from converting plasminogen to plasmin, mainly in the presence of fibrin. Plasmin, in turn, breaks fibrin into fibrin degradation products. Its fibrinolytic activity also induces consumption of coagulation factors, which later decreases the fibrinogen level and prolongs both prothrombin time (PT) and activated partial thromboplastin time (aPTT). These mechanisms result in intracranial bleeding [9, 10]. This process may explain why sICH patients in our study had aPTT prolongation and low fibrinogen levels.

HI is a common complication of acute ischemic stroke and is mostly linked to aICH. The extravasation of red blood cells is considered to be the cause of HI [11]. On the other hand, the PH is associated with sICH. Damaged arteries caused by ischemia and reperfusion are supposed to promote PH. However, PH is not common and usually engages with thrombolysis.

This study concurred with previous trials. We found that PH was significantly associated with sICH, whereas HI was associated with aICH.

The relationship between sICH and clinical outcomes is most consistent with PH especially in the PH-2 group according to ECASS classification. Contrarily, the relevant outcomes of HI or aICH are still unclear. In the ECASS II cohort, the PH-2 group had a nearly 50% of mortality [12]. In comparison to the no ICH and aICH groups, sICH patients had more unsatisfactory outcomes, including a higher rate of hematoma progression from follow-up imaging, a higher mRS, a higher NIHSS, a lower Barthel index, and a higher mortality rate at all study time points (at discharge, 14 days, and 90 days).

The presence of aICH also increases the risk of serious complications. At every time point of the study, the no ICH group had better outcomes than the aICH group. Unfavourable outcomes defined as mRS >2, NIHSS >15, Barthel index <25, and higher mortality rate were less frequently observed in no ICH group compared with aICH group. The patient with no ICH stayed in the hospital for a shorter time compared with the patients with aICH. This finding emphasized the importance of the occurrence of aICH. Even if patients did not have worsening neurological symptoms attributed to ICH, the clinical outcomes were consequently poor.

To date, there was only one retrospective cohort study that mentioned the relationship between treatment and the outcome of sICH. In that study, 5.2% of patients had sICH, and 42.2% of sICH had received reversal agents (mainly fresh frozen plasma and cryoprecipitate). The remaining obtained conservative treatment without a reversal agent. At 90-day follow-up, favourable clinical outcome (defined as mRS ≤2), rate of hematoma expansion and mortality showed no statistical difference in both the reversal agent group and conservative group [13]. Nevertheless, no current data regarding the management of aICH was reported. Data from multivariable analysis also demonstrated that atrial fibrillation, recurrent stroke, higher admission NIHSS, and lower Barthel index remained the potential risk factors for the unfavourable outcome (mRS >2 at 90 days). Based on this current study, aICH patients should be treated the same as those with sICH.

Multiple reversal agents are used in treating sICH cases. Cryoprecipitate is considered the potential agent as it contained fibrinogen, factor VIII, factor XIII, and von Willebrand factor. Its proposed mechanism is to correct hypofibrinogenemia and improve the intrinsic coagulation pathway. Platelet concentrate is also used in sICH. The theoretical mechanism is to correct the platelet inhibition from the tremendous production of D-dimers and glycoprotein IIb/IIIa from rtPA treatment. However, a small study was reported against the routine use of platelet concentration because of hematoma expansion [14]. The prothrombin complex concentrate (PCC) was found to be useful, but only for warfarin-related bleeding. Fresh frozen plasma (FFP) which contained nearly all coagulation factors revealed a controversial benefit due to its adverse events [13]. From our study, we found that reversal agents were primarily prescribed in the sICH group. Only 6% of patients with aICH received either FFP or cryoprecipitate. In our study, however, none of the reversal agents showed statistically significant results regarding 90-day mRS and other outcomes in both receiving and non-receiving groups. These findings were consistent with a previous study. Most sICH with reversal agents infusion showed unfavorable outcomes (mRS 4–6) compared with those who did not receive reversal agents (100% vs 82%, different in proportions 15.0, 95% CI -14.5 to 43.1) [13]. The role of the neurosurgical intervention was mainly considered when the hematoma expanded. We found that majority (60%) of patients with sICH underwent neurosurgery. We observed a low mortality rate in these patients after the surgery (45.1% vs 79.0%, P-value = 0.03).

The strength of this current study was that a large number of patients were included from a prospective stroke registry. The findings reflected real-world data on stroke treatment in a tertiary care center. The short-term and long-term outcomes of patients who experienced both

sICH and aICH were demonstrated. However, there were some limitations. First, no standard local practice guidelines for the management of rtPA-related bleeding were implemented in our institution. Therefore, there was heterogeneity in management. Second, we used the ECASS III and NINDS classifications to define sICH. This might bring a minor variation compared with other studies. We suggest a well-designed study to access the outcomes of individual reversal treatment.

## Conclusions

The incidence of rtPA associated with aICH is more common than sICH. Most of the reversal agents and neurosurgical interventions were used in symptomatic cases. The rtPA-associated aICH increased the risk of morbidity and mortality outcomes. Further treatment consensus, guideline generation, or clinical trials focusing on the treatment of rtPA-associated aICH may be required.

## Acknowledgments

The authors thank the principal investigators, research investigators, acute stroke unit care teams, and patients involved in this study.

## Author Contributions

**Conceptualization:** Kitti Thiankhaw.

**Data curation:** Chutithep Teekaput, Kanokkarn Teekaput, Chatree Chai-Adisaksopha.

**Formal analysis:** Chutithep Teekaput, Chatree Chai-Adisaksopha.

**Funding acquisition:** Chutithep Teekaput.

**Investigation:** Chutithep Teekaput, Kanokkarn Teekaput.

**Methodology:** Chutithep Teekaput, Kitti Thiankhaw, Chatree Chai-Adisaksopha.

**Project administration:** Chutithep Teekaput.

**Resources:** Chutithep Teekaput, Kitti Thiankhaw.

**Software:** Chutithep Teekaput.

**Supervision:** Surat Tanprawate, Chatree Chai-Adisaksopha.

**Validation:** Chutithep Teekaput, Kitti Thiankhaw, Kanokkarn Teekaput.

**Visualization:** Chutithep Teekaput, Surat Tanprawate.

**Writing – original draft:** Chutithep Teekaput.

**Writing – review & editing:** Chatree Chai-Adisaksopha.

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
