## [Decision Letter · Decision Letter 0]

22 Jul 2021

PONE-D-21-20195

Outcomes of Asymptomatic  Recombinant Tissue Plasminogen Activator Associated Intracranial Hemorrhage

PLOS ONE

Dear Dr. Teekaput,

Thank you for submitting your manuscript to PLOS ONE. After careful consideration, we feel that it has merit but does not fully meet PLOS ONE’s publication criteria as it currently stands. Therefore, we invite you to submit a revised version of the manuscript that addresses the points raised during the review process.

Please discuss the difference between patients with aICH and without ICH, especially regarding the risk factors.The statistical method needs to be revised.The data presentation in the table should be checked.

We look forward to receiving your revised manuscript.

Kind regards,

Quan Jiang, Ph,D.

Academic Editor

PLOS ONE

Journal Requirements:

 "No". 

"Funding: None"

"No"

"NO authors have competing interests". 

6. Please ensure that you refer to Figure 5 in your text as, if accepted, production will need this reference to link the reader to the figure.

Reviewers' comments:

Reviewer's Responses to Questions

**Comments to the Author**

1. Is the manuscript technically sound, and do the data support the conclusions?

Reviewer #1: Yes

Reviewer #2: Partly

2. Has the statistical analysis been performed appropriately and rigorously? 

Reviewer #1: Yes

Reviewer #2: No

3. Have the authors made all data underlying the findings in their manuscript fully available?

Reviewer #1: Yes

Reviewer #2: Yes

4. Is the manuscript presented in an intelligible fashion and written in standard English?

Reviewer #1: Yes

Reviewer #2: Yes

5. Review Comments to the Author

Reviewer #1: First of all, I would like to thank you for the article. In line 164-165, ‘’There were no 165 significant differences of the baseline characteristics of patients with regards to sex, age, comorbid diseases and prior medication use’’, How was this conclusion reached?Maybe the author should be more clear.

Thank you for your efforts.

Reviewer #2: The authors presented an issue regarding to asymptomatic tPA associated ICH which is less discussed in previous literatures. In this study, some different findings may propose that aICH may potentially lead to poor functional outcome, thus are worthy in further investigation. This may be the most important findings in this study. However, I have several comments.

1. I suggest to pay attention to discuss the difference between patients with aICH and without ICH, especially regarding the risk factors (past history, stroke classification, admission BP, sugar, stroke size, etc), potential predictors of aICH compared with patients without ICH, management (whether most patients with aICH did not need aggressive treatment) and outcome comparison. It may be the issue which is less discussed in previous literatures. So the data presentation may be considered to be revised.

2. I think the statistical method should be revised, especially regarding to the functional outcome comparison between groups. Some potential clinical factors should be put into multivariate analysis. From current manuscript, I can't confirm which factors have been put into multivariate analysis.

3. I think the data presentation in the table should be checked. For example, in Table 1, the overall patient number in the column of TOAST classification in patients without ICH (331+150+48+17) is fewer than the original number (559). Such same problem was also noted regarding the number in the column of prior medication.

6. PLOS authors have the option to publish the peer review history of their article (what does this mean?). If published, this will include your full peer review and any attached files.

Reviewer #1: No

Reviewer #2: No

---

## [Author Response · Author response to Decision Letter 0]

5 Oct 2021

Reviewer Comments, Responses, and Manuscript Changes

Reviewer 1

1. In lines 164-165, There were not 165 significant differences of the baseline characteristics of patients with regards to sex, age, comorbid diseases, and prior medication use, How was this conclusion reached?

Response: 

We thank the reviewer for the comment. We performed the univariable analyses comparing the differences among three groups (no ICH vs asymptomatic intracranial hemorrhage (aICH), no ICH vs symptomatic intracranial hemorrhage (sICH), and aICH vs sICH). When comparing three groups (using ANOVA or crosstab function), there was no statistical difference of variables. However, some variables had statistically significant differences between groups. Thus, we revised and added the information in Table 1.

(table 1 will be provided in the cover letter)

We revised the paragraph on page 8, line 163:

from “There were no significant differences of the baseline characteristics of patients with regards to sex, age, co-morbid diseases, and prior medication use.” 

To “Table 1 demonstrates baseline clinical characteristics of patients, classified into three groups (no ICH, aICH, and sICH). Atrial fibrillation and prior antiplatelet use were less frequently found in the no ICH group comparing to aICH and sICH groups. As compared to the no ICH group, the aICH and sICH groups had more unfavorable admission characteristics (higher mRS, higher NIHSS and lower Barthel index).”

2. Points should be after the references.

Response We revised the entire manuscript as suggested. 

3. Some English words should be checked. (125: past medical history )

Response We revised the sentence as suggested on page 6, line 125:

from “We collected demographic data, past medical history, laboratory data, and imaging data at stroke diagnosis.” 

to “We collected demographic data, medical history, laboratory data, and imaging data at stroke diagnosis.”

Reviewer 2

1. I suggest paying attention to discuss the difference between patients with aICH and without ICH, especially regarding the risk factors (history, stroke classification, admission BP, sugar, stroke size, etc), potential predictors of aICH compared with patients without ICH, management (whether most patients with aICH did not need aggressive treatment) and outcome comparison. It may be an issue that is less discussed in previous literature. So the data presentation may be considered to be revised.

Response: 

1) We thank the reviewer for these valuable comments. We revised Table 1. We compared the baseline characteristics of patients who did not have ICH and asymptomatic ICH. We performed the univariable analyses to compare the differences between the no ICH group and aICH group. Atrial fibrillation, prior antiplatelet use, and cardioembolic stroke subtype according to TOAST classification were less frequent in the no ICH group comparing to aICH group. On the other hand, the atherosclerotic stroke subtype was more frequently observed in the no ICH group than aICH group. Initial presentation of patients in the no ICH group was more favorable compared to aICH (lower admission NIHSS (median 9 [IQR 5, 15] vs 13 [IQR 8, 18], P-value =0.05) and higher Barthel index (median 25 [IQR 10, 55] vs 20 [IQR 5, 40], P-value =0.03)), respectively.

We revised the paragraph on page 8, line 163:

from “There were no significant differences of the baseline characteristics of patients with regards to sex, age, co-morbid diseases, and prior medication use.”

To “Table 1 demonstrates baseline clinical characteristics of patients, classified into three groups (no ICH, aICH, and sICH). Atrial fibrillation and prior antiplatelet use were less frequently found in the no ICH group comparing to aICH and sICH groups. As compared to the no ICH group, the aICH and sICH groups had more unfavorable admission characteristics (higher mRS, higher NIHSS and lower Barthel index).”

We added a new sentence on page 8, line 164: 

“Atrial fibrillation and prior antiplatelet used were less frequently found in no ICH group comparing to aICH and sICH groups.”

We added the paragraph on page 9, line 168:

“When comparing between patients who did not have ICH and those with aICH, the no ICH group had more atherosclerotic and less cardioembolic subtypes per the TOAST classification. Initial presentation of patients in no ICH group was more favorable compared to aICH (lower admission NIHSS (median 9 [IQR 5, 15] vs 13 [IQR 8, 18], P-value =0.05) and higher Barthel index (median 25 [IQR 10, 55] vs 20 [IQR 5, 40], P-value =0.03)), respectively.”

We revised Table 1.

Table 1. Baseline Characteristic of Patients Who Received Recombinant Tissue Plasminogen Activator for Acute Ischemic Stroke

(Table 1 will be provided in the cover letter)

2) Concerning predictors of aICH, we created Table 2 to demonstrate the results of a multivariable analysis of predictors for aICH. Cardioembolic stroke subtype was associated with an increased risk of aICH with an OR of 3.7 (95% CI 2.4-5.7). Moreover, admission NIHSS more than 15 was associated with the increment of aICH, OR 1.9 (95% CI 1.2-3.0)

We added the new topic on page 9, line 173:

“Risk factors of asymptomatic intracranial hemorrhage”

We added the paragraph on page 9, line 209 

“Table 2 demonstrates the results of multivariable analysis of risk factors for aICH. Cardioembolic stroke subtype was associated with an increased risk of aICH with an OR of 3.7 (95% CI 2.4-5.7). Moreover, admission NIHSS more than 15 was associated with the increment of aICH, OR 1.9 (95% CI 1.2-3.0)”

We added Table 2.

(Table 2 will be provided in the cover letter)

3) Concerning management, all patients who did not have ICH did not require blood transfusion or neurological intervention. 

We revised the paragraph on page 10, line 191:

“Table 4 demonstrates the management of rtPA-associated bleeding. The patients with aICH received any type of blood products less than the patients with sICH (7.8% vs 80.0%, P-value <0.001). Fresh frozen plasma was the most common blood product used, followed by cryoprecipitate. Packed red blood cell transfusion was given in 0.9% of the aICH group and 32.0% in the sICH group (P-value <0.001). Craniotomy was performed in 0.9% of the aICH group and 60.0% in the sICH group (P-value <0.001).”

We added the paragraph on page 10, line 197:

“Focusing on patients without ICH, transfusion of blood product or neurosurgical intervention was not required in all patients.”

4) Concerning the outcomes, the patients without ICH had more favorable outcomes comparing with the patients with aICH and sICH (higher proportion of mRS >2, NIHSS >15, Barthel <25, and higher mortality rate at day 14 and day 90) as shown in Figure 2, 3 and 4. We analyzed the association of length of stay (LOS) vs no ICH, aICH and sICH groups. We found that the no ICH group had a shorter length of stay (LOS) comparing with aICH group and sICH group (median 5 days (IQR 3, 9), 10 days (IQR 5, 20) and 9 days (IQR 5,15), P-value < 0.001), respectively.

We added the paragraph on page 11, line 208:

“Furthermore, the no ICH group had a shorter length of stay (LOS) comparing with aICH group and sICH group (median 5 days [IQR 3, 9],10 days [IQR 5, 20] and 9 days [IQR 5, 15], P-value <0.001), respectively.”

We revised the paragraph on page 11, line 216:

“NIHSS and Barthel index at 14 and 90 days after stroke diagnosis are shown in Figures 3 and 4. A higher proportion of patients with sICH had the worst stroke outcomes (defined as NIHSS >15 or Barthel index <25) compared to those with no ICH and aICH at day 14 (P-value =0.003 and P-value <0.001, respectively), and at day 90 (P-value <0.001 and P-value <0.001, respectively).”

We added the new topic on page 11, line 221:

“Mortality”

We added the paragraph on page 12, line 292:

“Patients in the no ICH group had a significantly lower proportion of all-cause mortality at 14 days and 90 days as compared to patients with aICH and sICH. The Kaplan-Meier curve of overall survival is demonstrated in Figure 5. When compared to no ICH patients, aICH and sICH were associated with an increased risk of 90-day mortality adjusted by age, HR 3.7, 95% CI 1.6-8.9, and HR 40.8, 95% CI 20.7-80.6, respectively.”

We revised Figure 5.

Figure 5 will be provided in the cover letter and attached Figure.

2. I think the statistical method should be revised, especially regarding the functional outcome comparison between groups. Some potential clinical factors should be put into multivariate analysis. From the current manuscript, I can't confirm which factors have been put into multivariate analysis.

Response:

1.) We performed the univariable and multivariable analyses to evaluate the risk factors of the poor outcome as defined by mRS >2. Table 5 demonstrated the independent risk factors for poor stroke outcomes. After adjusting for confounders (age, atrial fibrillation, serum albumin level, admission NIHSS, and Barthel index), aICH and sICH increased risk of unfavorable stroke outcome. 

We added the paragraph on page 10, line 203:

“Table 5 demonstrated multivariable analysis of unfavorable mRS at 90 days. After adjusting for confounders (age, atrial fibrillation, serum albumin level, admission NIHSS, and Barthel index), aICH and sICH increased risk of unfavorable stroke outcome when compared to patients without ICH with OR 2.3 (95% CI 1.5-3.6), P-value <0.001, and OR 21.8 (95% CI 5.1-92.4), P-value <0.001, respectively.”

We added Table 5

Table 5 Multivariable Analysis of Unfavorable Outcomes Defined as Modified Rankin Scale > 2 in Patients Who Had Acute Ischemic Stroke and Received rtPA.

(Table 5 will be provided in the cover letter)

2.) Concerning the all-cause mortality. We performed a univariable analysis using the Cox-Proportional hazard model. Table 1R was for review propose only. We found that intracranial hemorrhage was the only clinical risk factor that was independently associated with death. We decided to include age in the final Cox-Proportional hazard model due to the biological plausibility that age might contribute to mortality. Table 1R (for review only) demonstrates the hazard ratios for all-cause mortality in the final model that included ICH status and age. Consequently, we reported the hazard ratios of all-cause mortality in the manuscript page 11, line 225, after adjusting for age. 

Table 1R. (for review) Cox-Proportional Hazard Model for All-Cause Mortality

(Table 1R will be provided in the cover letter)

3. I think the data presented in the table should be checked. For example, in Table 1, the overall patient number in the column of TOAST classification in patients without ICH (331+150+48+17) is fewer than the original number (559). Such same problem was also noted regarding the number in the column of prior medication.

Response:

We revised the data in Table 1 as demonstrated above.

1) In TOAST classification, we added the “undetermined” row. 

2) We corrected the percentage of unfavorable outcomes in the no ICH group. On page 10 line 202, we changed from “40.5%” to “25.9%”.

---

## [Decision Letter · Decision Letter 1]

4 May 2022

PONE-D-21-20195R1Outcomes of Asymptomatic  Recombinant Tissue Plasminogen Activator Associated Intracranial HemorrhagePLOS ONE

Dear Dr. Chai-adisaksopha,

Thank you for submitting your manuscript to PLOS ONE. After careful consideration, we feel that it has merit but does not fully meet PLOS ONE’s publication criteria as it currently stands. Therefore, we invite you to submit a revised version of the manuscript that addresses the points raised during the review process.

The manuscript has been assessed by one reviewer, and his comments are appended below.

The reviewer has raised a number of concerns that need attention. He feels that the conclusion should be better addressed and he has also requested to revise the Results section.  He also has concerns about the language used and request copy editing.

Could you please carefully revise the manuscript to address all comments raised?

We look forward to receiving your revised manuscript.

Kind regards,

Lorena Verduci

Staff Editor

PLOS ONE

Journal Requirements:

Reviewers' comments:

Reviewer's Responses to Questions

**Comments to the Author**

1. If the authors have adequately addressed your comments raised in a previous round of review and you feel that this manuscript is now acceptable for publication, you may indicate that here to bypass the “Comments to the Author” section, enter your conflict of interest statement in the “Confidential to Editor” section, and submit your "Accept" recommendation.

Reviewer #2: All comments have been addressed

2. Is the manuscript technically sound, and do the data support the conclusions?

Reviewer #2: Partly

3. Has the statistical analysis been performed appropriately and rigorously? 

Reviewer #2: Yes

4. Have the authors made all data underlying the findings in their manuscript fully available?

Reviewer #2: Yes

5. Is the manuscript presented in an intelligible fashion and written in standard English?

Reviewer #2: No

6. Review Comments to the Author

Reviewer #2: After reading the revised manuscript, most of my previous comments have been addressed. I still have several comments:

1. For Line 140: Remove the comma before mRS.

“Functional outcomes, including, mRS, NIHSS, and Barthel Index were collect at the time of stroke diagnosis, 14 days, and 90 days post-rtPA.”

2. For Line 142: Remove the “a” before 90-day mRS

“A primary outcome of this study was a 90-day mRS. “

3. For Line 177: Substitute ICH to sICH

“When comparing between patients who did not have ICH and those with aICH.”

4. For Line 174-176 : As compared to the no ICH group, the aICH and sICH groups had more unfavorable admission characteristics (higher mRS, higher NIHSS and lower Barthel index).

The meaning is similar to line 179-181 “Initial presentation of patients in no ICH group was more favorable compared to aICH (lower admission NIHSS (median 9 [IQR 5, 15] vs 13 [IQR 8, 18], P-value =0.05) and higher Barthel index (median 25 [IQR 10, 55] vs 20 [IQR 5, 40], P-value =0.03)), “. Please combine these two parts.

5. After line 204, description regarding that bleeding progression is more common in sICH group may be added.

6. For Line 216: may add “group” after sICH.

“Figure 2 demonstrated the 90-day mRS among no sICH, aICH, and sICH.”

7. For section after Line 214: I suggest to combine the “The 90-day modified rankin scales” with the “Outcomes of rtPA associated intracranial hemorrhage”and “Mortality”.

8. May revise the conclusion in the abstract and manuscript. May consider the below description: “rtPA associated aICH still increased the risk of morbidity and mortality outcomes. Further treatment consensus or guideline generation or clinical trials focusing on the treatment of rtPA-associated aICH may be needed”.

9. Please consider English editing for the grammar again.

7. PLOS authors have the option to publish the peer review history of their article (what does this mean?). If published, this will include your full peer review and any attached files.

Reviewer #2: No

---

## [Author Response · Author response to Decision Letter 1]

31 May 2022

Reviewer Comments, Responses, and Manuscript Changes

Reviewer 2

1. For Line 140: Remove the comma before mRS.

“Functional outcomes, including, mRS, NIHSS, and Barthel Index were collect at the time of stroke diagnosis, 14 days, and 90 days post-rtPA.”

Modification: We remove the comma as the reviewer suggested.

See line 327 page 7

2. For Line 142: Remove the “a” before 90-day mRS

“A primary outcome of this study was a 90-day mRS.”

Modification: We remove the “a” as the reviewer suggested.

See line 329 page 7

3. For Line 177: Substitute ICH to sICH

“When comparing between patients who did not have ICH and those with aICH.”

Modification: We deleted the sentences “When comparing between patients who did not have ICH and those with aICH, the no ICH group had more atherosclerotic and less cardioembolic subtypes per the TOAST classification. Initial presentation of patients in no ICH group was more favorable compared to aICH (lower admission NIHSS (median 9 [IQR 5, 15] vs 13 [IQR 8, 18], P-value =0.05) and higher Barthel index (median 25 [IQR 10, 55] vs 20 [IQR 5, 40], P-value =0.03)), respectively.”

These sentences were forementioned in the previous paragraph. 

See line 385 page 8

4. For Line 174-176 : As compared to the no ICH group, the aICH and sICH groups had more unfavorable admission characteristics (higher mRS, higher NIHSS and lower Barthel index).

The meaning is similar to line 179-181 “Initial presentation of patients in no ICH group was more favorable compared to aICH (lower admission NIHSS (median 9 [IQR 5, 15] vs 13 [IQR 8, 18], P-value =0.05) and higher Barthel index (median 25 [IQR 10, 55] vs 20 [IQR 5, 40], P-value =0.03)), “. Please combine these two parts.

Modification: We combined the two paragraph. We deleted “Initial presentation of patients in no ICH group was more favorable compared to aICH (lower admission NIHSS (median 9 [IQR 5, 15] vs 13 [IQR 8, 18], P-value =0.05) and higher Barthel index (median 25 [IQR 10, 55] vs 20 [IQR 5, 40], P-value =0.03))”

See line 308 page 8

5. After line 204, description regarding that bleeding progression is more common in sICH group may be added.

Modification: We added the sentence “According to the lower incidence of bleeding progression in the aICH group, patients with aICH received fewer blood products than patients with sICH (7.8% vs 80.0 percent, P-value <0.001).”

See line 431 page 9

6. For Line 216: may add “group” after sICH.

“Figure 2 demonstrated the 90-day mRS among no sICH, aICH, and sICH.”

Modification: We added the word “group.”

See line 515 page 10

7. For section after Line 214: I suggest to combine the “The 90-day modified rankin scales” with the “Outcomes of rtPA associated intracranial hemorrhage”and “Mortality”.

Modification: We combine the “The 90-day modified rankin scales” with the “Outcomes of rtPA associated intracranial hemorrhage”and “Mortality”.

See line 526 page 10

8. May revise the conclusion in the abstract and manuscript. May consider the below description: “rtPA associated aICH still increased the risk of morbidity and mortality outcomes. Further treatment consensus or guideline generation or clinical trials focusing on the treatment of rtPA-associated aICH may be needed”.

Modification: We modified the sentences from “Majority of aICH patients were left untreated. Further clinical trials focusing on the treatment of rtPA-associated aICH are urgently needed in oder to improve clinical outcomes in these patients.” 

to 

“The rtPA associated aICH increased the risk of morbidity and mortality outcomes. Further treatment consensus, guideline generation, or clinical trials focusing on the treatment of rtPA-associated aICH may be required.”

See line 83 page 4 and line 989 page 15

9. Please consider English editing for the grammar again.

Comments: The manuscript was revised and language edited by English speaking person. 

 

Other modification

1. In the introduction, we added the sentence “Although numerous research has focused on the long-term outcomes of sICH, there is insufficient evidence on the long-term outcomes of asymptomatic ICH (aICH). ”

See line 120 page 5

2. In the introduction, we change the sentence “This study aimed to investigate the incidence, risk factors, and characteristics of aICH secondary from thrombolytic therapy in patients who had an acute ischemic stroke. We sought to determine reversal strategies, surgical intervention, and the outcomes.” to

“The objective of this study is to determine the incidence, risk factors, and characteristics of aICH in stroke patients after receiving thrombolytic therapy. We also determine the reversal strategies, surgical intervention, and the outcomes.”

See line 124 page 5

3. In method, we change the paragraph from “Data were retrieved from The Chiang Mai University Hospital Stroke Registry between 1995 and 2019. This registry prospectively collected consecutive patients who were diagnosed with all types of acute stroke. The clinical data, laboratory, imaging, and outcomes were systematically collected. Patients with acute ischemic stroke with onset less than 3-4.5 hours, without contraindications to rtPA administration (e.g. history of previous intracranial hemorrhage or abnormal laboratory, etc.) would be given rtPA 0.9 mg/kg, maximum dose 90 mg over 60 minutes intravenously within 60 minutes after arrival at an emergency department. After that, all patients were admitted to the Acute Stroke Unit (ASU) with vital signs and neurological outcomes monitored, including the National Institute of Health Stroke Scale (NIHSS), modified Rankin scale (mRS), and Barthel index. Twenty-four hours after rtPA infusion, patients subsequently underwent a follow-up computed tomography (CT) scan of the brain to detect radiologic evidence of intracranial hemorrhage. In case of worsening of neurological symptoms, defined as an increase in NIHSS ≥ 4, patients underwent emergency brain imaging. If intracranial bleeding was detected, the comprehensive ASU care team, consisting of neurologists, neurosurgeons, rehabilitation physicians, nurses, and pharmacists evaluated and classified patients into a symptomatic or asymptomatic intracranial hemorrhage. Neurosurgical intervention or reversal agents would be given according to the comprehensive ASU care team consensus. When patients were discharged from the ASU, NIHSS, mRS, Barthel index and all-cause mortality were recorded. The NIHSS, mRS, Barthel index, and all-cause mortality would be collected again at the 14-day and 90-day follow-up visits.” to 

“Data were retrieved from the Chiang Mai University Hospital Stroke Registry between 1995 and 2019. This registry prospectively collected consecutive patients who were diagnosed with all types of acute stroke. All stroke patients who were 18 years or older, diagnosed with acute ischemic stroke with onset less than 3-4.5 hours, without contraindications to rtPA administration (e.g. history of previous intracranial hemorrhage or abnormal laboratory, etc.) would be given rtPA 0.9 mg/kg, maximum dose 90 mg over 60 minutes intravenously within 60 minutes after arrival at an emergency department. The clinical data, laboratory, imaging, and outcomes were systematically collected. All patients were subsequently admitted to the Acute Stroke Unit (ASU). Vital signs and neurological outcomes including the National Institute of Health Stroke Scale (NIHSS), modified Rankin scale (mRS), and Barthel index were monitored. All patients underwent a follow-up computed tomography (CT) scan of the brain 24 hours after receiving rtPA to detect the radiologic signs of intracranial bleeding. In case of worsening neurological symptoms, defined as an increase of NIHSS ≥ 4, the patients would undergo emergency brain imaging. If intracranial bleeding was detected, the comprehensive ASU care team comprised of neurologists, neurosurgeons, rehabilitation physicians, nurses, and pharmacists would evaluate and classify patients into symptomatic or asymptomatic intracranial hemorrhage. Neurosurgical intervention or reversal agents would be given according to the comprehensive ASU care team consensus. When patients were discharged from the ASU, NIHSS, mRS, Barthel index and all-cause mortality were recorded. The NIHSS, mRS, Barthel index, and all-cause mortality would be re-collected at the 14-day and 90-day follow-up visits. Patients who did not have a follow-up brain imaging or who had missing data on functional outcomes at any time point were excluded from the study.”

See line 128 page 5

---

## [Editor Report · Decision Letter 2]

18 Jul 2022

Outcomes of Asymptomatic Recombinant Tissue Plasminogen Activator Associated Intracranial Hemorrhage

PONE-D-21-20195R2

Dear Dr. Chai-adisaksopha,

We’re pleased to inform you that your manuscript has been judged scientifically suitable for publication and will be formally accepted for publication once it meets all outstanding technical requirements.

Kind regards,

James Mockridge

Staff Editor

PLOS ONE

---

## [Editor Report · Acceptance letter]

21 Jul 2022

PONE-D-21-20195R2 

Outcomes of Asymptomatic Recombinant Tissue Plasminogen Activator Associated Intracranial Hemorrhage 

Dear Dr. Chai-adisaksopha:

I'm pleased to inform you that your manuscript has been deemed suitable for publication in PLOS ONE. Congratulations! Your manuscript is now with our production department. 

Kind regards, 

on behalf of

Dr James Mockridge 

Staff Editor

PLOS ONE